# Mechanisms of Antioxidant Resistance in Different Wheat Genotypes under Salt Stress and Hypoxia

**DOI:** 10.3390/ijms242316878

**Published:** 2023-11-28

**Authors:** Neonila V. Kononenko, Elena M. Lazareva, Larisa I. Fedoreyeva

**Affiliations:** 1All-Russia Research Institute of Agricultural Biotechnology, Timiryazevskaya 42, 127550 Moscow, Russia; nilava@mail.ru (N.V.K.); lazareva_e@yandex.ru (E.M.L.); 2Biological Department, M.V. Lomonosov Moscow State University, Leninskie Gory 1, 119991 Moscow, Russia

**Keywords:** AOS, ROS, abiotic stress, apoptosis-like, *Triticum aestivum* L., *Triticum durum* Desf

## Abstract

Various stressors lead to an increase in ROS and damage to plant tissues. Plants have a powerful antioxidant system (AOS), which allows them to neutralize excess ROS. We detected an intense fluorescent glow of ROS in the cells of the cap, meristem, and elongation zones in the roots of wheat *Triticum aestivum* (Orenburgskaya 22 variety) and *Triticum durum* (Zolotaya variety). An increase in ROS was accompanied by DNA breaks in the nuclei of wheat root cells, the release of cytochrome c from mitochondria into the cytoplasm, and the translocation of phosphatidylserine into the outer layer of the plasma membrane under salt stress and hypoxia. The different resistances of the two wheat varieties to different abiotic stresses were revealed. The soft wheat variety Orenburgskaya 22 showed high resistance to salt stress but sensitivity to hypoxia, and the durum wheat variety Zolotaya showed tolerance to hypoxia but high sensitivity to salt stress. Different activations of AOS components (GSH, MnSOD, Cu/ZnSOD, CAT, PX, GPX, and GST) were revealed in different wheat genotypes. The basis for the tolerance of the Zolotaya variety to hypoxia is the high content of glutathione (GSH) and the activation of glutathione-dependent enzymes. One of the mechanisms of high resistance to salt stress in the Orenburgskaya 22 variety is a decrease in the level of ROS as a result of the increased activity of the *MnSOD* and *Cu*/*ZnSOD* genes. Identifying the mechanisms of plant tolerance to abiotic stress is the most important task for improving breeding varieties of agricultural plants and increasing their yield.

## 1. Introduction

Wheat is the most common type of agricultural crop [1]. In the process of wheat growth, stress factors damage plants, slow down growth, and reduce productivity [2,3]. Plants quickly adapt to environmental changes. Depending on the mechanisms and speed of adaptation, plants can be classified as tolerant or sensitive varieties to abiotic stresses. O_2_ is an essential element for plant survival. Oxygen deficiency, which is formed as a result of heavy rainfall, interferes with respiration and other biochemical processes [4]. Hypoxia associated with the waterlogging and flooding of roots affects the productivity of plants in ecosystems [5,6]. Plants often experience physiological hypoxia due to limited diffusion of O_2_ or its rapid consumption [7,8,9,10]. In addition, some other conditions can also lead to hypoxia, for example, salt stress, which is one of the most common and studied abiotic factors, leads to a decrease in cell permeability for O_2_ [11,12].

In flooded organs and tissues, the O_2_ concentration decreases. Regardless of the conditions that cause low O_2_ levels, plants respond to hypoxic stress in two alternative ways: accelerating shoot growth to achieve normal O_2_ levels, and dormancy, which slows down growth and conserves metabolic resources [13,14]. They differ in phytohormone signaling, which determines the degree of tolerance to hypoxia [15]. In flood-resistant plants, aerenchyma formation has been observed [15,16], Similarly, adventitious root formation can also improve O_2_ levels in plants under waterlogged conditions [16]. Meanwhile, the production of ROS, as well as an increase in antioxidant activity, increases the tolerance of plants to hypoxia [17].

An increase in ROS in plants occurs under the influence of various stress conditions, such as hypoxia, drought, and salinity [18]. By maintaining the optimal concentration of ROS, the antioxidant system serves to protect plants from toxicity [19,20]. ROS are also signaling molecules that regulate the growth, development, and response of plants to the environment [21].

Stress factors lead to damage that causes an increase in the formation of ROS. A significant increase in ROS is called oxidative stress. The level of formation of highly toxic ROS in plant cells is controlled by antioxidants [22,23,24]. It has been shown that ROS are involved in the processes of proliferation and differentiation, as well as in programmed cell death (PCD). It has been shown that the electron transport chain contains auto-oxidizable enzymes that convert O_2_ to O_2_^−^ [25,26], while the PSII chain can produce superoxide under intense light illumination [27]. ROS are also generated by mitochondria during oxidative phosphorylation reactions in the electron transport chain [28,29]. Mitochondrial changes lead to the release of mitochondrial intermembrane proteins, the disruption of the electron transport chain, changes in the transmembrane potential difference, and the formation and inhibition of ROS [30]. An increase in lipid oxidation in mitochondrial membranes violates their integrity and leads to swelling and lysis of mitochondria. This disrupts the energy supply of cells and reduces their adaptive ability to stress factors.

Plants have developed a complex antioxidant system that suppresses the accumulation of ROS and consists of a complex of enzymes (superoxide dismutase (SOD); catalase (CAT); peroxidase (PX); glutathione reductase (GR); monodehydroascorbate reductase (MDHAR); dehydroascorbate reductase (DHAR); glutathione peroxidase (GPX); guaicol peroxidase (GOPX); and glutathione-S-transferase, (GTS)) and a complex of low-molecular-weight compounds (glutathione (GSH), phenolic compounds, alkaloids, non-protein amino acids, and α-tocopherols). These systems work together to regulate uncontrolled oxidative cascades and protect cells from stress [31,32]. It is logical to assume that increased plant resistance is associated with the presence of a powerful antioxidant system in the body [33]. Superoxide dismutase (SOD), which catalyzes the transformation of O_2_^-^ radical anions into H_2_O_2_, is the main enzyme of the antioxidant system involved in most physiological and biochemical processes [34,35]. Enzymes are divided into three types depending on the metal cofactor: FeSODs, MnSODs, and Cu/ZnSODs [36,37,38]. Various SOD isoforms differ in amino acid sequences, crystal structures, and subcellular localizations and have different sensitivities to H_2_O_2_ in vitro [37,38,39,40]. Cu/ZnSODs are located in chloroplasts, the cytoplasm, and/or extracellular space [41], and MnSODs are found in plant mitochondria [42,43]. FeSODs are localized in chloroplasts and the plant cytoplasm [44].

Soil salinization is the most common abiotic stress, as a result of which the yield of many agricultural crops, primarily wheat, is significantly deteriorated. Plants under salt stress respond by reducing physiological functions, which can lead to deviations from normal plant development [45]. In those plants that have not developed adaptation mechanisms and have weak stress resistance mechanisms, significant tissue damage occurs, which leads to the death of the entire plant. Tissue damage is accompanied by changes in ROS levels. An increase in the activity of all components of the antioxidant system (AOS) is one of the conditions for plant tolerance to any abiotic stress. Determining resistant wheat varieties to one or another stress effect is a priority for many researchers. Previously, we considered the identification of tolerant wheat varieties to salt stress. Different resistances to sodium chloride were found in the Orenburgskaya 22 and Zolotaya wheat varieties [46].

In this study, we focused on studying the formation and localization of ROS in wheat *T. aestivum* (Orenburgskaya 22) and *T. durum* (Zolotaya) as a result of exposure to abiotic stresses such as saline and hypoxia, identifying the active components of AOS in two wheat varieties under the influence of stress factors. It was planned to examine in detail the mechanisms of tolerance of two different genotypes of wheat and to identify differences in the process of autophagy and PSD in different varieties of wheat under the influence of different stressors.

## 2. Results

### 2.1. Morphometric Parameters

In our experiment, the effects of 150 mM NaCl and hypoxia on the antioxidant systems of two varieties of soft wheat, Orenburgskaya 22 and durum Zolotaya, were examined. Wheat was grown in rolls. Table 1 shows the morphometric parameters of plant changes in response to any stress conditions for 6-day seedlings of the two wheat varieties after exposure to two stress variants.

Inhibition of root and shoot growth as a result of exposure to sodium chloride was observed in both wheat varieties, especially in the Zolotaya variety. In the Oreburgskaya 22 variety, the root length and shoot height decreased by 11 and 29%, respectively, and in the Zolotaya variety, by 16 and 40%, respectively. Under hypoxic conditions, the length of the roots of Orenburgskaya 22 practically did not change, and the height of the shoots decreased slightly by 9%. In the Zolotaya variety, hypoxia also led to a slight decrease in shoot height by 8%, but the length of the roots increased by 17%. Thus, according to morphometric indicators, the Zolotaya variety has resistance to hypoxia.

### 2.2. Chlorophyll Content

Table 2 shows data on the contents of Chl a and b in the shoots of the Orenburgskaya 22 and Zolotaya wheat varieties grown under different conditions.

It was found that the contents of Chl a and Chl b in the Zolotaya variety were significantly lower than those in the Orenburgskaya 22 variety (by 1.67 and 1.92 times, respectively). The table data show that any stress, sodium chloride or hypoxia, led to a decrease in the contents of Chl a and Chl b in the shoots of Orenburgskaya 22 wheat, and these changes were more significant than in the Zolotaya variety. Salt exposure led to a decrease in the contents of Chl a and Chl b in the Orenburgskaya 22 variety by 2.09 and 2.38 times, respectively, while hypoxia did by 3.3 and 3.67 times, respectively. Salt stress also had a negative effect on the shoots of the Zolotaya variety, but it was less significant than that on the shoots of the Orenburgskaya 22 variety. The contents of Chl a and Chl b decreased by 1.84 and 1.87 times, respectively. Despite the fact that the level of the drops in Chl a and Chl b contents under salt stress in the Orenburgskaya 22 variety was more significant than in the Zolotaya variety, the Chl a and Chl b contents remained higher compared with the Zolotaya variety. It is interesting to note that hypoxia led to a slight increase in the contents of Chl a and Chl b in the shoots of Zolotaya wheat variety compared with the control by 1.09 and 1.04 times, respectively. An increase in the contents of Chl a and Chl b under hypoxia in the Zolotaya variety and a sharp drop in the levels of Chl a and Chl b in the Orenburgskaya 22 variety resulted in their contents in the Zolotaya variety becoming two times higher than in the Orenburgskaya 22 variety.

It should be noted that the Chl a/Chl b ratio in the Zolotaya variety was higher than in the Orenburgskaya 22 variety, although the contents of Chl a and Chl b in the Oreburgskaya variety were higher than in the Zolotaya variety. When Orenburgskaya 22 wheat was treated with stress factors, the Chl a/Chl b ratio increased relative to the control. In contrast with the Orenburgskaya 22 variety, the Zolotaya variety exhibited an inverse relationship, whereby the Chl a/Chl b ratio decreased.

### 2.3. ROS under Stress

#### 2.3.1. ROS under Salt Stress

Our studies using the ROS marker carboxy-H2DFF found that under salinity with 150 mM NaCl, ROS production was visualized in all root tissues, but with different staining intensities in cells from different root zones. Root zones with varying degrees of fluorescence were also noted. The data obtained show that under salinity, the most intense ROS staining was observed in the cap and meristem zones, which indicates an increase in the level of ROS production (Figure 1). Under salt stress, an increase in ROS was also observed in the cells of the epidermis and cortex and, to a lesser extent, in the cells of the central cylinder (Figure 1). With an increase in the NaCl concentration in the solution, an increase in the fluorescence intensity in the roots of the Orenburgskaya 22 variety to 30% was observed, while at the same time, in the Zolotaya variety, the ROS production increased to 55%. Therefore, under oxidative stress caused by salinity, ROS are preferentially localized in epidermal and cortical cells in the areas of the cap and division. Moreover, the Orenburgskaya 22 variety resulted in being the most resistant to salinity compared with the Zolotaya variety.

#### 2.3.2. ROS under Hypoxia

The amount of ROS production was determined by the fluorescence intensity in the root cells of the Orenburgskaya 22 and Zolotaya varieties under the action of hypoxia for 24 h. Although ROS production increased in the roots of the Zolotaya variety under hypoxia compared with the control variant, compared with the Orenburgskaya 22 variety, the intensity of ROS fluorescence was lower: 25% and 41%, respectively (Figure 1). The root zones were noted from the most intense fluorescence to its absence. The data obtained show that intense ROS staining was observed in the roots in the cap and meristem zones (Figure 1A,B). Compared with the control, the increase in ROS levels was observed to the greatest extent in the cells of the epidermis and cortex and, to a lesser extent, in the cells of the central cylinder, i.e., similar to the effect of salt stress (Figure 1A,B).

### 2.4. ROS Trigger Apoptosis-like Cell Death

In the salt-tolerant variety Orenburgskaya 22, DNA breaks were noted early in 0.4% of the nuclei of control cells, and in the presence of NaCl, in 19%. In the salt-sensitive variety Zolotaya, DNA breaks were observed in 0.5% of control cell nuclei and in 32% in the presence of NaCl [46].

DNA breaks in the nuclei of root cells were also found in response to hypoxia (Figure 2a′,b′,c). In the Orenburgskaya 22 variety, DNA breaks were found in 11% of root cell nuclei, and in the Zolotaya variety, in 8%. The root tissue cells of the Orenburgskaya 22 variety were larger than those of the Zolotaya variety. Most of the cells were in the G2 period of interphase and prophase of mitosis. Binuclear cells were observed, which indicated a violation of cytokinesis, the cytoplasm of which was vacuolized. An increase in the number of nuclei with DNA breaks indicates that hypoxia is an inducer of cell death.

Interestingly, under the influence of hypoxia, phosphatidylserine was found in the outer layer of the plasma membrane in 8% of cells in the Orenburgskaya 22 variety and only in 3% in the Zolotaya variety, while the cell nuclei were stained with propidium iodide, which indicates their necrotic death (Figure 3).

Cytochrome c was immunodetected in the mitochondria of wheat root tissue cells (Figure 4). Abiotic stress led to an increase in the permeability of the outer mitochondrial membrane and the release of cytochrome c into the cytoplasm. In the salt-tolerant variety Orenburgskaya 22 in the presence of NaCl, cytochrome c was found in the cytoplasm in 41% of the cells, and in the sodium-chloride-sensitive variety Zolotaya, in 54% of cells.

### 2.5. Antioxidant System

#### 2.5.1. H_2_O_2_ Content

The H_2_O_2_ contents in the control shoots and roots of both wheat varieties were close in value (Figure 5). Salt stress and hypoxia had different effects on the H_2_O_2_ content in different wheat varieties. Only a slight accumulation of H_2_O_2_ occurred in the roots as a result of stress factors in the Orenburgskaya 22 variety, whereas in the shoots, H_2_O_2_ increased 1.4 times under the action of NaCl and 1.6 times under hypoxia. It should be noted that NaCl had the most negative effect on the roots of Zolotaya wheat, causing the greatest increase in the formation of H_2_O_2_ (two times higher compared with the control). Hypoxia also caused an increase in the H_2_O_2_ content in the roots of the Zolotaya variety, but only by 1.5 times. The accumulation of H_2_O_2_ in the Zolotaya wheat variety was also observed in the shoots, by 1.6 times under the action of NaCl and by 1.1 times under hypoxia. Thus, under salt stress, the greatest accumulation of H_2_O_2_ was observed in the roots of the Zolotaya variety, and under hypoxia, H_2_O_2_ accumulated in the shoots of the Orenburgskaya 22 variety.

#### 2.5.2. Antioxidant Activity

Since free radical oxidation is a chain of branched reactions initiated by various types of ROS, during which various degradation products of molecules with their own activity are formed, there is no universal method for assessing the antioxidant activity (AOA) of biologically active substances. Moreover, the AOA value strongly depends on the extraction method. The results obtained with the help of only one test should be interpreted with great caution in relation to biological objects. Therefore, in vitro, AOA is currently assessed using several test systems. In our study, two test systems were used, which were based on different methods of extraction of the materials under study: water and alcohol. (R)-4-[1-Hydroxy-2-(methylamino)ethyl]-benzene-1,2-diol (HMAEB) and 2,2-diphenyl-1-picrylhydrazyl (DPPH) were used as substrates.

The AOA value in both wheat varieties was approximately the same and only slightly differed: in the roots of the Orenburgskaya 22 variety, it was 1.09 times higher than in the Zolotaya variety, and in the shoots of the Zolotaya variety, it was only 1.03 times higher than in Orenburgskaya 22 (Table 3). Abiotic stress was accompanied by a decrease in AOA compared with the control variants. Under the influence of NaCl on the Orenburgskaya 22 wheat variety, a slight decrease in the AOA value was observed, while in the roots of the Zolotaya variety, the action of salt led to a drop in the AOA by 2.8 times. In the shoots of the Zolotaya variety, salt stress led to a two-fold drop in AOA. In contrast with the effect of NaCl, hypoxia did not lead to such catastrophic consequences for the Zolotaya variety, although the AOA decreased in both the roots and shoots. In contrast with NaCl, hypoxia had a more significant effect on the roots and shoots of the Orenburgskaya 22 variety (by 1.5 and 1.55 times, respectively) than on the Zolotaya variety.

The ARA values were much higher than the AOA values, but the changes had a similar trend to those in the AOA values. This was probably due to the fact that the alcohol extraction of plant material promotes the extraction of low-molecular-weight antioxidants, such as polyphenols and flavonoids.

#### 2.5.3. Glutathione Content

It was determined that the GSH content in the roots of both wheat varieties was significantly higher than in shoots: in the Orenburgskaya 22 variety by 3.63 times, and in the Zolotaya variety, by 4.47 times (Figure 6). The Zolotaya and Orenburgskaya 22 wheat varieties differed significantly in GSH contents in both the roots and shoots. The GSH content in the shoots of the Zolotaya variety was 1.42 times higher compared with the Orenburgskaya 22 variety, and in the roots, it was 1.75 times higher.

Abiotic stress was accompanied by a significant increase in the GSH in both the shoots and roots of the Zolotaya variety. In the shoots of this wheat variety, the GSH content increased by 3.85 times under salt stress and 7.68 times under hypoxia. In the roots of the Zolotaya variety, an increase in the GSH content was also observed by 1.59 times with the addition of NaCl and by 2.58 times under hypoxia. It should be noted that under hypoxia in the roots of the Zolotaya variety, the GSH content reached high values of −5.42 mM. Unlike the Zolotaya variety, in the shoots of the Orenburgskaya 22 variety, abiotic stresses did not lead to an increase in the GSH content (1.2 times under the influence of NaCl and hypoxia). A more significant increase (1.64 times) in the GSH content occurred in the roots of Orenburgskaya 22 under hypoxia.

#### 2.5.4. Expression of Genes

The response of the antioxidant system to abiotic stresses in plants can be considered as both a change in the level of enzymatic activity and a change in the level of expression of genes encoding enzymes of the antioxidant system. In our study, were studied the expressions of the *MnSOD*, *Cu*/*ZnSOD*, *PX*, *GPX*, *GST*, and *CAT* genes. The results are resented in Figure 7.

The expression level of the *MnSOD* gene in the roots of the Zolotaya variety was 2.4 times lower than in the Orenburgskaya 22 variety. Interestingly, in the shoots, this value was only 1.09 times lower. The presence of sodium chloride in the medium led to a slight decrease in the expression level of the *MnSOD* gene in the roots of the Orenburgskaya 22 wheat variety by 1.19 times and remained unchanged in the shoots. There was a slight decrease in the expression level of the mitochondrial gene *MnSOD* in the Zolotaya variety by 1.16 times under the influence of NaCl in both the roots and shoots. Hypoxia caused a sharp increase in the expression of the *MnSOD* gene in the Orenburgskaya 22 variety: in the roots, by 1.74 times, and in the shoots, by 1.6 times. In the Zolotaya variety, an increase in the expression level of the *MnSOD* gene under hypoxia was also observed, but only in the roots, and this increase was much more modest, by only 1.15 times.

The level of expression of the chloroplast gene *Cu*/*ZnSOD* in the Orenburgskaya 22 variety also exceeded the level of this gene in the Zolotaya variety: in the roots, by 2.23 times, and in the shoots, by 1.5 times. The presence of NaCl had virtually no effect on the expression level of *Cu*/*ZnSOD* in both the Orenburgskaya 22 variety and the Zolotaya variety. Hypoxia had a significant effect on the expression of *Cu*/*ZnSOD*, especially in the Zolotaya variety: in the roots, the relative expression level increased by 5.01 times, and in the shoots, by 2.64 times. In the Orenburgskaya 22 variety, hypoxia was accompanied by an increase in *Cu*/*ZnSOD* expression only in the roots by 1.6 times, while in the shoots, it remained unchanged. This difference in the expression levels of *Cu*/*ZnSOD* in different wheat varieties under hypoxia led to the level of *Cu*/*ZnSOD* expression in the roots of the Zolotaya variety becoming 1.41 times higher compared with the Orenburgskaya 22 variety.

Although the GST enzyme is an important element in ROS detoxification, the level of expression of the *GST* gene in the Orenburgskaya 22 variety was low and remained virtually unchanged under all growing conditions. However, in the Zolotaya variety, significant regulation of *GST* gene expression was observed under salt stress and especially under hypoxia, both in the roots and shoots. In the presence of NaCl in the roots, the level of *GST* gene expression increased by 1.1 times, while in the shoots, it increased by 1.35 times. The opposite picture was observed under hypoxia: in the roots, the level of *GST* gene expression increased by 5.02 times, and in the shoots, by 2.63 times.

The expression level of the *GPX* gene in both wheat varieties was high. In the Zolotaya variety, it exceeded that in the Orenburgskaya 22 variety: in the roots by 1.24 times, and in the shoots, by 2.2 times. An increase in the NaCl concentration led to a decrease in the level of *GPX* expression in the Zolotaya variety in both the roots and shoots. Hypoxia also led to a decrease in the level of *GPX* expression, but only in the roots; in the shoots, its level increased by 1.18 times. It is interesting to note that in the Orenburgskaya 22 variety, salt stress and hypoxia led to a decrease in the level of *GPX* expression in the roots. In the shoots of this wheat variety, there was an increase in the expression of the GPX gene in the presence of NaCl by 1.56 times and a decrease in the expression of this gene under hypoxia by 1.42 times.

In the Orenburgskaya 22 variety, the expression level of the *PX* gene was high in both the roots and shoots. Hypoxia did not lead to significant changes in the level of *PX* gene expression in this wheat variety. However, it should be noted that the addition of NaCl to the medium was accompanied by a 1.89-fold decrease in the level of expression of the PX gene in the Orenburgskaya 22 variety in the roots and a 1.52-fold decrease in the shoots. A different picture was observed in the Zolotaya variety. In the control root samples of this wheat variety, the expression level was 5.93 times lower than in Orenburgskaya 22, and in the shoots, it was 1.85 times lower. In the shoots of the Zolotaya variety, neither salt stress nor hypoxia affected the expression of the *PX* gene, whereas in the roots, stress led to an increase in the expression of the *PX* gene, under NaCl treatment, by 3.34 times, and under hypoxia, by 5.53 times.

The expression level of the *PX* gene in the Zolotaya variety in the shoots was two times lower than in the Orenburgskaya 22 variety and seven times lower in the roots.

It is interesting to note that *CAT* expression was more active in the roots of the Zolotaya variety (2.54 times) than in those of the Orenburgskaya 22 variety, and in the shoots, *CAT* expression was most active in the Orenburgskaya 22 variety (1.33 times). The level of *CAT* expression in the shoots of the Orenburgskaya 22 variety was 4.55 times higher than in the roots. Hypoxia did not affect the expression of *CAT* in the shoots of the Orenburgskaya 22 variety, but it affected its expression in the roots, and the level dropped even lower, by 1.43 times. An increase in the NaCl concentration led to an increase in *CAT* expression in the Orenburgskaya 22 variety in the roots by 1.96 times, and in the shoots, by 1.34 times. Exposure to hypoxia and salt stress was accompanied by an increase in the level of *CAT* expression in the Zolotaya variety in both the roots and shoots. It should be noted that hypoxia caused a more significant increase in *CAT* expression compared with NaCl: in the roots, by 1.59 and 1.47 times, respectively, and in the shoots, by 1.41 and 1.3 times, respectively.

## 3. Discussion

ROS are important signaling molecules and are vital for cellular metabolism [47]. Under the influence of stressors, the production of ROS in cells increases, and reactions to neutralize excess ROS begin. Any adverse effect on the plant that causes a block in metabolism, also affecting growth and development, can be considered as stress for the plant.

Excessive production of ROS under oxidative stress in hypoxia is part of stressful situations. A decreased oxygen content increases the likelihood of ROS formation by inhibiting the mitochondrial electron transport chain [48]. In the transition from normal to hypoxia, there is an increase in the formation of ROS, and the mitochondria undergo swelling, releasing Ca^2+^ and cytochrome c. The content of H_2_O_2_ in the apoplast and plasma membrane in wheat root cells under hypoxia was determined [49]. In contrast with the control, wheat mitochondria under hypoxia are characterized by an increase in volume, a decrease in matrix density, and the breakdown of cristae associated with deep swelling [50]. In order to survive, an organism under conditions of hypoxic stress induces PCD [51].

ROS production was visualized in all root tissues but with different staining intensities. Root zones with varying degrees of fluorescence were also noted. Under conditions of abiotic salinity stress, the most intense ROS staining was observed in the zone of the cap and meristem. An increase in the content of ROS was observed in epidermal and cortical cells and, to a lesser extent, in the cells of the central cylinder. Thus, under salt stress and hypoxia, ROS were predominantly localized in the cells of the epidermis and cortex in the areas of the cap and division.

The accumulation of ROS in the cells and tissues of wheat seedling roots under the influence of oxidative stress indicates a violation of ROS homeostasis. As a result of exposure to abiotic stresses, ROS accumulated in the roots of both wheat varieties; however, the level of ROS in the Zolotaya variety exceeded the ROS in the Orenburgskaya 22 variety by 20–35%, depending on the stress factor (Figure 1).

It is characteristic that in both control wheat varieties, the roots and shoots contained almost the same amount of H_2_O_2_. This fact is probably important for the normal development of plants. Salt stress and hypoxia have different effects on different wheat genotypes, which was reflected in the accumulation of H_2_O_2_ in the roots and shoots. There was no accumulation of excess H_2_O_2_ in the roots of the Orenburgskaya 22 variety. At the same time, in the shoots of the Orenburgskaya 22 variety, the amount of H_2_O_2_ increased under the influence of both NaCl and hypoxia. This fact indicates that the AOS in roots and shoots has some differences. These differences may be due to both the different compositions of AOS components and their activities. The effects of salt stress and hypoxia on the Zolotaya variety were accompanied by an increase in the H_2_O_2_ content in both the roots and shoots. The Zolotaya variety showed sensitivity to the action of NaCl, and the greatest accumulation of peroxide occurred in the roots of the Zolotaya variety. Thus, the two wheat varieties showed a significant difference in the accumulation of H_2_O_2_ in the roots and shoots as a result of salt stress and hypoxia. This suggests that different AOA mechanisms exist in roots and shoots to remove excess ROS in different wheat genotypes.

Salt stress reduces the content of chlorophyll (Chl) and carotenoids as well as metabolic functions in the cell [45,52]. Osmotic stress associated with ionic imbalance as well as hypoxia can lead to oxidative damage [53,54]. With a high concentration of salts in the soil, damage to the photosynthetic apparatus occurs [55], causing changes in the redox balance [41]. Analysis of the contents of chlorophylls a and b is an excellent tool for quantifying damage to the photosynthetic apparatus caused by abiotic stress [56,57,58]. Plant stress adversely affects photosynthetic electron transfer in PSI and PSII and chlorophyll biosynthesis [59]. Therefore, the photosynthetic apparatus is a sensor of stressors, which is responsible for the imbalance of cellular energy [60].

The contents of Chl a (4.13 mg) and Chl b (1.69 mg) in the control shoots of the Orenburgskaya 22 variety were significantly higher than those of the Zolotaya variety (2.47 mg and 0.88 mg, respectively) (Table 2). It was found that any stress, sodium chloride or hypoxia, led to a significant decrease in the contents of Chl a and Chl b in the shoots of the Orenburgskaya 22 variety. In the Zolotaya variety, the contents of Chl a and Chl b also decreased in the presence of NaCl. However, this drop was not as catastrophic as that of the Orenburgskaya 22 variety. Moreover, as a result of the effect of hypoxia on the shoots of the Zolotaya variety, an increase in the content of both chlorophylls was observed. An increase in the contents of Chl a and Chl b under hypoxia in the Zolotaya variety and a sharp drop in the levels of Chl a and Chl b in the Orenburgskaya 22 variety led to the fact that their contents in the Zolotaya variety became two times higher than in the Orenburgskaya 22 variety. It can be assumed that the photosynthetic apparatus of the Zolotaya variety is more stable to stress influences than that of the Orenburgskaya 22 variety. Based on this assumption, the high content of chlorophylls in the control shoots of the Orenburgskaya 22 variety becomes understandable.

Increased ROS formation triggers the antioxidant defense of wheat. In the process of evolution, plants have developed a powerful antioxidant system to protect against external influences. This system includes both an enzymatic complex and low-molecular-weight antioxidants. The antioxidant system protects plants from damage caused by oxidative stress. SOD activity was shown to be elevated due to salt stress, suggesting a link between increased ROS production and a defense mechanism that reduces stress-induced oxidative damage. H_2_O_2_ is a product of SOD activity and is reactive and harmful to cells and must be eliminated by conversion to H_2_O [34,35]. In plants, enzymes that regulate cellular H_2_O_2_ levels are CAT (catalase) and GPX (glutathione peroxidase) [61,62,63]. CAT activity modulates the amounts of O_2_ and H_2_O_2_ and reduces the risk of forming OH^-^ radicals that are highly reactive [64]. Glutathione (GSH) is a low-molecular-weight component of the AOS involved in H_2_O_2_ detoxification [32].

The AOA value in the Orenburgskaya 22 wheat variety in the roots was slightly higher than that in the Zolotaya variety, and in the shoots of the Zolotaya 22 variety, it was higher than in the Orenburgskaya 22 variety (Table 3). When the Orenburgskaya 22 wheat variety was exposed to NaCl, a slight decrease in AOA values occurred. In contrast with the Orenburgskaya 22 variety, a significant decrease in the level of total AOA (2.8 times) was observed in the roots of the Zolotaya variety. In contrast with NaCl, hypoxia only slightly reduced the AOA values in both the roots and shoots of the Zolotaya variety and significantly reduced them in the Orenburgskaya 22 variety (1.5 times).

Glutathione is a tripeptide (L-glutamyl-L-cysteinylglycine (GSH)). The protective effect of glutathione is accompanied by its oxidation sulfhydryl group and conversion to glutathione disulfide (GSSG). It is believed that H_2_O_2_ detoxification with the participation of glutathione can occur in two ways. The first is the reduction of H_2_O_2_ by glutathione in a reaction catalyzed by glutathione peroxidase. The second way to reduce hydrogen peroxide is associated with the oxidation of ascorbic acid to dehydroascorbate under the action of ascorbate peroxidase. Glutathione is used to detoxify lipid peroxides using glutathione-S-transferase. A high GSH/GSSG ratio and the activity of enzymes associated with glutathione metabolism are markers of resistance to stressors [65]. Chloroplasts are considered the main organelles not only for the synthesis of glutathione but also for its localization in plant cells. At the same time, glutathione can also be synthesized and localized in the cytoplasm [61]. Glutathione has also been found in rather high concentrations in mitochondria, although the possibility of its synthesis in these organelles has not yet been proven [62].

The Zolotaya and Orenburgskaya 22 wheat varieties differ significantly in GSH content in both roots and shoots. In the roots, the GSH content in the Zolotaya variety (2.43 mM) was almost two times higher than that in the Orenburgskaya 22 variety (1.32 mM) (Figure 6). In the shoots of both wheat varieties, the GSH content was almost two times lower than in the roots. In the shoots of the Orenburgskaya 22 variety, abiotic stresses did not lead to a significant increase in the GSH content (1.2 times under the influence of NaCl and hypoxia). A more significant increase (1.64 times) in GSH content occurred in the roots of Orenburgskaya 22 under hypoxic conditions. Unlike the Orenburgskaya 22 variety, in the Zolotaya variety, under the influence of factors, especially hypoxia, the GSH content increased, especially in the shoots: under salt stress, by 3.85 times, and under hypoxia, by 7.68 times (Figure 6). Under hypoxic conditions in the roots of the Zolotaya variety, the GSH content reached high values of 5.42 mM. Such a significant difference in the GSH contents in the different wheat varieties indicates different mechanisms of antioxidant protection and its most significant participation in the Zolotaya variety.

In the control wheat samples, the activities of the *MnSOD* and *Cu*/*ZnSOD* genes in the Orenburgskaya 22 variety were significantly higher than those in the Zolotaya variety, both in the roots and shoots. The expression levels of the *MnSOD* and *Cu*/*ZnSOD* genes in the Orenburgskaya 22 variety were almost two times higher than the expression levels in the Zolotaya variety (Figure 7). Although salt stress either did not affect the expression levels of the *MnSOD* and *Cu*/*ZnSOD* genes in both varieties or led to a slight decrease in them, in the Orenburgskaya 22 variety, the expression levels remained higher than in the Zolotaya variety. Under the influence of hypoxia, an almost two-fold increase in *MnSOD* expression was observed in the Orenburgskaya 22 variety and remained almost virtually unchanged in the Zolotaya variety. However, the expression level of *Cu*/*ZnSOD* increased by 5.01 times in the roots of the Zolotaya variety after exposure to hypoxia and by 2.64 times in the shoots. Based on these indicators, it was assumed that the expressions of the *MnSOD* and *Cu*/*ZnSOD* genes in the Orenburgskaya 22 variety, especially in the roots, are more important for ROS detoxification than in the Zolotaya variety. The accumulation of excess H_2_O_2_ under salt stress does not occur in the Orenburgskaya 22 variety, which can be explained by the high levels of expression of the *MnSOD* and *Cu*/*ZnSOD* genes. The hypoxia was accompanied by an increase in the H_2_O_2_ content, which led to a significant increase in the levels of expression of *MnSOD* and *Cu*/*ZnSOD* in the Orenburgskaya 22 variety. However, the expression levels of the *MnSOD* and *Cu*/*ZnSOD* genes in the Zolotaya variety were lower than those in the Orenburgskaya 22 variety, which probably indicates the activation of other components of the AOS to remove excess ROS under stress, so it should be noted that under hypoxia, the level of expression of the chloroplast *Cu*/*ZnSOD* gene significantly increased.

Enzymes such as peroxidase (PX), catalase (CAT), glutathione peroxidase (GPX), and glutathione-S-transferase (GST) are also active participants in the AOS. The main function of PX is the neutralization of hydrogen peroxide with the help of various reducing agents (often phenolic compounds). However, along with antioxidant activity, peroxidases can exhibit oxidase activity when transferring electrons from reducing agents (for example, NADH) to oxygen and the formation of ROS [63]. It is interesting to note that the level of PX expression decreased almost two-fold in the roots of the Orenburgskaya 22 variety under salt stress. At the same time, the H_2_O_2_ content in the roots of Orenburgskaya 22 did not change. It can be assumed that in this wheat variety, PX could also participate in the oxidative process and the formation of ROS, and a decrease in the expression of *PX* had a beneficial effect on the H_2_O_2_ content. In the roots of the Zolotaya variety, an increase in *PX* expression was observed both under salt stress and especially under hypoxia. In this variant of wheat, PX was probably involved mainly in the removal of H_2_O_2_. Although the H_2_O_2_ content in the roots of the Zolotaya variety increased by almost 2 times and the level of *PX* expression increased by 3.34 times, the level of *PX* expression remained low compared with the Orenburgskaya 22 variety. This conclusion was also confirmed by the data on hypoxia. The H_2_O_2_ content in the roots of the Zolotaya variety under hypoxia increased by 1.5 times, and the level of *PX* expression increased by 5.53 times, i.e., the more active the *PX*, the lower the H_2_O_2_ content in the Zolotaya variety.

The main function of CAT is the decomposition of H_2_O_2_ into water and molecular oxygen. The enzymes are localized mainly in peroxisomes and glyoxisomes. Its specific form was also found in mitochondria [66,67]. It is believed that CAT, unlike PX, works effectively at high concentrations of H_2_O_2_ and is capable of converting approximately six million H_2_O_2_ molecules into H_2_O and O_2_ in 1 min [67]. *CAT* expression was more active in the roots of the Zolotaya variety than in those of the Orenburgskaya 22 variety. Exposure to hypoxia and salt stress was accompanied by an increase in the level of *CAT* expression in the Zolotaya variety in both the roots and shoots. It should be noted that hypoxia caused a more significant increase in *CAT* expression compared with NaCl. It can be assumed that CAT is one of the main players in the process of H_2_O_2_ neutralization under salt stress and hypoxia. An increase in the NaCl concentration led to an increase in *CAT* expression in the Orenburgskaya 22 variety in the roots by 1.96 times, and in the shoots, by 1.34 times. Although the level of *CAT* expression significantly increased under salt stress in the roots of the Orenburgskaya 22 variety, its activity was low, and it can be assumed that the role of the main participant in the removal of excess ROS belongs to SOD enzymes.

GPX is the main component of the ascorbate–glutathione cycle and maintains a reduced pool of GSH [22]. The level of *GPX* expression in the Zolotaya variety exceeded the level in the Orenburgskaya 22 variety, especially in the roots. This fact seems to be associated with the GSH content. GSH is known to be a substrate for the GPX enzyme. The content of GSH, which is a substrate for the GPX enzyme, in the roots of the Zolotaya variety was 1.75 times higher than that in the Orenburgskaya 22 variety, and it was 1.42 times higher in the shoots. An increase in the concentration of NaCl led to a decrease in the level of *GPX* expression in the roots of the Orenburgskaya 22 variety and to an almost two-fold increase in the shoots. Although the content of GSH in Zolotaya increased under the influence of NaCl, the level of *GPX* expression decreased in the shoots as well as in the roots. Under hypoxic conditions, the level of *GPX* expression remained at a high level in the Zolotaya variety. GST is responsible for detoxification and is able to conjugate GSH for elimination from the plant [68]. The level of *GST* expression in all variants of growing conditions in both wheat varieties was low and remained unchanged after exposure to abiotic stresses, except for the impact of hypoxia on the Zolotaya variety both in the roots and shoots.

It is likely that the detoxification process in the Zolotaya variety under hypoxia occurred with the participation of both GST and GPX. The high levels of *GST* and *GPX*, as well as the high content of GSH, in the Zolotaya variety determined the tolerance of this variety to hypoxia. A decrease in the level of *GPX* expression, a low level of *GST* activity, and a low GSH content in the Orenburgskaya 22 variety led to its high sensitivity to hypoxia.

Based on the results obtained, a scheme is presented for the activation of enzymatic and non-enzymatic AOS (glutathione) under the influence of stress factors in the roots and shoots of wheat of different genotypes (Figure 8). The results obtained reveal different levels of activation of various AOS components in hexaploid and tetraploid wheat. The protective reaction of AOS in the tetraploid wheat variety Zolotaya to the negative effects of hypoxia was due to the high content of GSH and activation of glutathione-dependent enzymes. The hexaploid wheat variety Orenburgskaya 22 showed high tolerance to salt stress, and the mechanism of this resistance was associated with high levels of expression of the *MnSOD* and *Cu*/*ZnSOD* genes.

An increase in the content of H_2_O_2_ is accompanied by damage to the cell tissue in plants. Plants either acclimatize to the negative effects of stress conditions or trigger one or another variant of PCD [69]. There are two ways of PCD development in plants: “apoptosis-like” cell death, such that its markers are DNA breaks, cytochrome c release from mitochondria, and phosphatidylserine transfer to the outer layer of the membrane, and “vacuolar death”, in which large vacuoles and autophagosomes are formed [70].

Abiotic and biotic stresses have been shown to cause the degradation of intracellular components during autophagy [71]. An increase in ROS generation under stress promotes the formation of autophagosomes, which attenuate oxidative stress [72]. It was previously established that autophagy is induced in the cells of the wheat roots of the Orenburgskaya 22 and Zolotaya varieties under the action of salt [46]. Under the action of abiotic stresses, the autophagy process was observed after a change in the permeability of the outer mitochondrial membrane (Figure 3). Vacuolization of the cell cytoplasm in the roots of the Zolotaya wheat variety was detected in most cells under hypoxic conditions. Only 8% of cells after exposure to flooding died by programmed necrosis. Vacuolization of the cytoplasm is probably an element of cell protection from hypoxia in the Zolotaya variety.

Our data show that reliable markers of cell death as a result of abiotic stresses are the formation of DNA breaks (Figure 2) and the release of cytochrome c from mitochondria (Figure 4). The maximum number of DNA breaks was formed after salt exposure, and damage to the outer mitochondrial membrane was observed under flooding. At the same time, the number of nuclei with breaks in the salt-sensitive variety Zolotaya was two times higher than in the salt-tolerant variety Orenburgskaya 22 (Figure 2). However, in the salt-sensitive variety Zolotaya under hypoxia, the number of nuclei with breaks was lower than in the Orenburgskaya 22 variety. Using markers, it was shown that under the action of accumulated ROS, a process similar to apoptosis is triggered. DNA breaks were found in the nuclei of root cells, and the release of cytochrome c into the cytoplasm indicated the mitochondrial pathway of cell death under stress (Figure 2, Figure 3 and Figure 4).

Plants are constantly exposed to various environmental influences. These impacts can be both long-term and short-term and vary in intensity. When exposed to stress factors, plants either acclimatize or adapt to them. Depending on the mechanisms triggered by plants in response to stress, it is possible to determine their tolerance to stress, rapid acclimatization, and, finally, adaptation to the stress factor. In the absence of certain mechanisms of tolerance in a plant to a certain stress, its duration, and high intensity, serious cell damage occurs in the plant, which ultimately leads to the death of the plant. According to Lichtenbohl’s theory, two types of stress, stress (eu-stress) or mild stress, can activate a plant’s adaptation mechanisms to this stress factor, and this is beneficial for the plant. Another type of stress, distress, is negative for the plant and is accompanied by slower growth and significant morphological changes and often leads to the death of the plant [73].

In our study, two types of stress factors were used: 150 mM NaCl and hypoxia, and the effects on wheat of these factors were short-term, within 24 h. Based on morphometric indicators, it follows that the two wheat varieties differ in adaptive properties to these stress factors. It was noted that the soft wheat variety Orenburgskaya 22 was more resistant to 150 mM NaCl than the durum wheat variety Zolotaya, but it was less resistant to hypoxia compared with the Zolotaya variety. At the same time, the Zolotaya variety quickly adapted to hypoxia; these conditions were favorable for it. The Zolotaya variety exhibited a significant elongation of the main roots and a change in the architecture of the root system. It has been shown that the impact of stress factors on wheat is accompanied by damage to cellular tissues, leading to autophagy via the mitochondrial pathway and necrosis. The Orenburgskaya 22 variety quickly adapted to salt stress by turning on tolerance mechanisms, while practically no necrotic cells were noted in the roots of this variety. Hypoxia for the Orenburgskaya 22 variety was distress. In this wheat variety, there are no mechanisms for rapid adaptation, which is accompanied by an increase in the percentage of cell death via the necrotic path. The opposite picture is observed in the Zolotaya variety. This variety exhibits tolerance to hypoxia and sensitivity to NaCl, which is manifested in an increased number of necrotic cells.

## 4. Materials and Methods

### 4.1. Plants

Two varieties of wheat, Orenburgskaya 22 (*Triticum aestivum* L., 2n = 42) and Zolotaya (*Triticum durum* Desf., 2n = 28), developed by the Orenburg Research Institute of Agriculture of the steppe ecological group (FGBNU Federal Scientific Center of the Russian Academy of Sciences, Orenburg, Russia), were used in this study. The effects of 150 mM NaCl and hypoxia on the antioxidant systems of the two wheat varieties were studied in this experiment. To study the effect of sodium chloride, wheat rolls after 5 days were transferred to a solution of 150 mM NaCl for 24 h. To study hypoxia, wheat roots were completely immersed in water for 24 h [74]. Each experiment was carried out with three repetitions.

### 4.2. Analysis of Chlorophyll Content

Chlorophyll was extracted via 80% acetone extraction [75]. Chlorophyll (Chl) levels were quantified according to the absorbance values at 665 nm and 649 nm, which were measured spectrophotometrically with a nanophotometer IMPLEN. The contents of Chl a and b were determined using the following formulas:chl a = [(11.63 × A_665_) − (2.39 × A_649_)] × Vml/1000 × Wmg
chl b = [(20.11 × A_649_) − (5.18 × A_665_)] × Vml/1000 × Wmg
where Vml is the total volume of the extract, and Wmg is the sample of plant.

Data were expressed as means ± standard deviation (SD; n = 30), and significant differences were determined as *p* < 0.05.

### 4.3. Fluorescence Microscopy

To determine ROS using the fluorescence method, root tips (4–5 mm) of 6-day-old seedlings were incubated in 25–50 nM of carboxy-H2DFFDA (Thermo Fisher Scientific, Waltham, MA, USA) according to our method [74]. The samples were analyzed under an Olympus BX51 fluorescent microscope (Olympus corporation, Tokyo, Japan) with a 10X objective at a wavelength of 490 nm. Images were obtained using a Color View digital camera (Munster, Germany).

### 4.4. Apoptosis Detection Assay

To identify root cells at the stages of programmed cell death (PCD), phosphatidylserine was detected using the Xpert Annexin V-FITC Apoptosis Detection Assay (Grisp, Maspalomas, Spain). The root tips of 6-day-old seedlings (5 mm) were macerated with 1% cellulase (Sigma Aldrich, St. Louis, MO, USA), then transferred to an Annexin V-FITC solution, incubated for 30 min, washed, transferred to a propidium iodide solution, and placed in Mowiol U-88 (Hoechst, Frankfurt am Main, Germany) with the addition of DAPI (4,6-diamidino-2-phenylindole) (Sigma Aldrich, USA) according to a previously described method [46].

### 4.5. TUNEL Analysis

Nuclear DNA breaks were detected using the TUNEL method. The preparations were placed in cocodelate buffer with deoxynucleotide diltransferase (Silex, Moscow, Russia), 3′-labeled probes with dATP (Silex, Russia), and fluorescein (Silex, Russia). The reaction was stopped with an SSC solution. After washing, the preparations were placed in Mowiol U-88 (Hoechst) with the addition of DAPI [46].

### 4.6. Cytochrome c Detection

Immunocytochemical determination of cytochrome c was carried out according to a previously described method [46]. The preparations were incubated with rabbit antibodies to cytochrome c, then washed and incubated with goat anti-rabbit antibodies (Sigma, USA), stained with DAPI, and placed in Mowiol U-88 (Hoechst, Frankfurt am Main, Germany). The preparations were analyzed with an Axiovert 200 M microscope (Zeiss, Jena, Germany). Images were taken with an AxioCam HRm camera (Zeiss, Jena, Germany).

### 4.7. Biochemical Analysis

Antioxidant activity (AOA) was determined by blocking the process of oxidation of a 0.1% aqueous solution of (R)-4-[1-hydroxy-2-(methylamino)ethyl]-benzene-1,2-diol (HMAEB). The absorbance was measured at λ = 347 nm. AOA was calculated using the formula: (A-Ao/A) × 100% [76]. The antiradical activity (ARA) was determined by the decrease in the coloration of the 5 × 10^−5^ M alcohol solution of 2,2-diphenyl-1-picrylhydrazyl (DPPH). The absorbance was measured at λ = 517 nm. ARA was calculated using the following formula: (Ao-A/Ao) × 100% [77]. The concentration of peroxide in aqueous solutions of plant material was determined by the reduction in the coloration of a 0.02 M solution of KMnO_4_. The absorbance was measured at λ = 480 nm [78]. The glutathione (GSH) content in mM was determined using the Elman method by the appearance of color after the addition of a 0.01 M alcohol solution. The absorption absorbance was measured at λ = 412 nm [79].

### 4.8. Total RNA Isolation and Gene Expression Analysis

Using a standard RNA isolation kit-Extran RNA Syntol (Moscow, Russia), RNAs were isolated from wheat roots and shoots grown under different conditions. cDNAs were synthesized via reverse transcription according to the standard method (Syntol, Moscow, Russia). The cDNA concentration was determined spectrophotometrically with an IMPLEN nanophotometer [46].

RT-PCR using SYBR Green I (Syntol) was performed in a CFX 96 Real-Time thermal cycler (BioRad, Hercules, CA, USA). Information on the structure of the *Cu*/*ZnSOD*, *MnSOD*, *Per*, *GPX*, *CAT,* and *GST* genes in wheat was obtained from NCBI. Primers for the genes were synthesized with Syntol (Appendix A). Each RT-PCR reaction was performed with three repeats. *GaPDh* was used as a reference gene.

### 4.9. Statistical Methods

The statistical processing of the experimental data was carried out using analysis of variance using the ANOVA program and Student’s *t*-test (R version 4.3.1.) with significant differences determined at *p* < 0.05. The least significant difference method was used to test significance. Values are presented as the means ± standard deviations of triplicate biological replicates.

## 5. Conclusions

Under natural conditions, plants are exposed to stress that can inhibit their growth and development. One of the visible symptoms of abiotic stress is impaired seedling growth. The Orenburgskaya 22 and Zolotaya wheat varieties resulted in being good models for studying the influence of abiotic stresses. The Orenburgskaya 22 variety showed high resistance to 150 mM NaCl and high sensitivity to flooding. The Zolotaya variety, on the contrary, was resistant to hypoxia and unstable in high NaCl concentrations. Stress factors lead to an increase in ROS in plants and increased cell damage. It should be noted that NaCl had the most negative effect on the roots of Zolotaya wheat, causing the greatest formation of H_2_O_2_. Greater accumulation of H_2_O_2_ in shoots was observed in the Orenburgskaya 22 wheat variety under hypoxic conditions. The AOS was activated to remove the excess negative radicals in the wheat. Different AOS components were activated in the different wheat varieties. For the soft wheat variety Orenburgskaya 22, the most important AOS components were *MnSOD* and *Cu*/*ZnSOD*, and for the durum wheat variety Zolotaya, they were GSH and the GSH-dependent enzymes GST and GPX. Therefore, different mechanisms of resistance in the different wheat varieties to salt stress and hypoxia have been identified. Different wheat adaptation mechanisms resulted in different types of cellular damage.

Upon using PCD markers in cells of the Orenburgskaya 22 variety, no transfer of phosphatidylserine to the cell surfaces under salt stress was observed, in contrast with the Zolotaya variety. However, DNA breaks were detected in the nuclei and metaphase chromosomes, as well as the release of cytochrome c into the cytoplasm, which indicates the mitochondrial pathway of death for some root cells under salinity. We observed similar death markers in a larger number of cells in the Orenburgskaya 22 variety under hypoxia.

A comprehensive characterization of the response of wheat to abiotic stress is an important and necessary tool for identifying the mechanisms of plant tolerance. The results obtained are of practical value for the use of tolerant varieties of wheat in areas with negative effects of various environmental factors on plants.

## Figures and Tables

**Figure 1 ijms-24-16878-f001:**
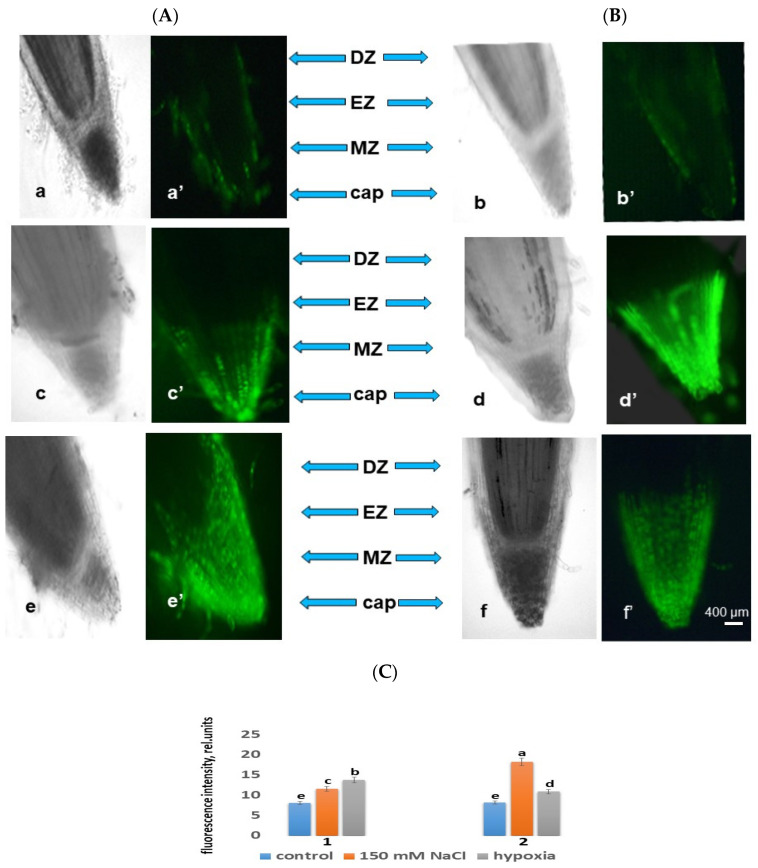
Distributions of ROS+ and ROS in cells in root zones of 6-day seedlings of Orenburgskaya 22 (**A**) and Zolotaya (**B**) wheat varieties: (**a**,**a′**,**b**,**b′**)—control; (**c**,**c′**,**d**,**d′**)—150 mM NaCl; and (**e**,**e′**,**f**,**f′**)—hypoxia. DZ—differentiation zone, EZ—elongation zone, and MZ—meristem zone. Bar: 400 µm. (**C**) The intensity of ROS fluorescence under the influence of stress factors in wheat: 1—Orenburgskaya 22 variety; 2—Zolotaya variety. a–e—indicate significant difference were determined (*p* < 0.05).

**Figure 2 ijms-24-16878-f002:**
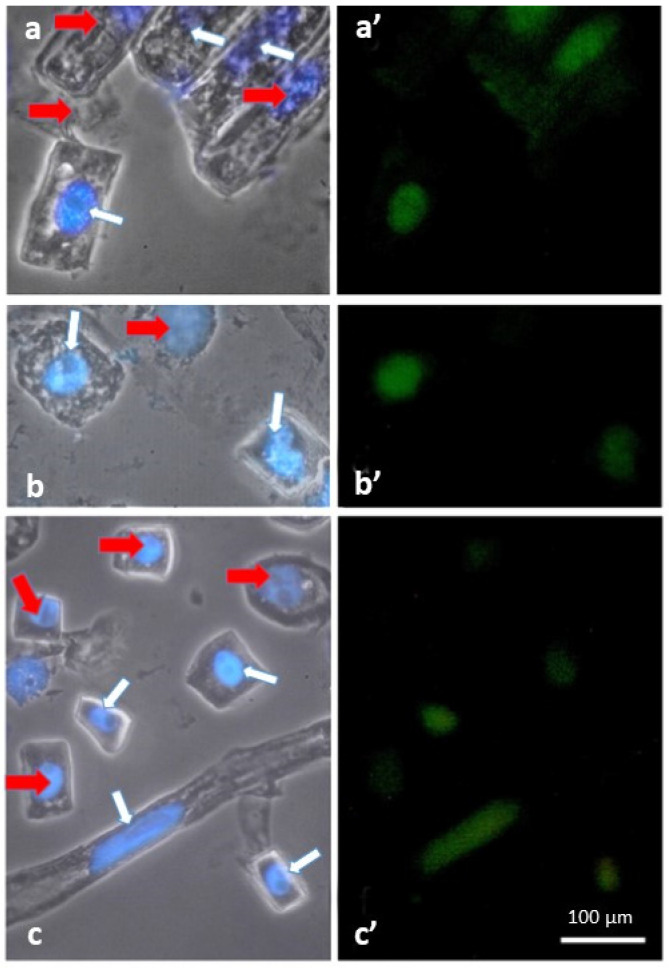
DNA breaks in the nuclei of root cells of 6-day-old seedlings of *T. aestivum* variety Orenburgskaya 22 and *T. durum* variety Zolotaya detected using the TUNEL method. Phase contrast of (**a**,**b**) Orenburgskaya 22 and (**c**) Zolotaya cells after exposure to hypoxia for 24 h. White arrows mark the nuclei of cells with 3′-end DNA breaks, and red arrows mark nuclei without them. (**a′**,**b′**,**c′**) Nuclei with DNA breaks (green). Bar: 100 μm.

**Figure 3 ijms-24-16878-f003:**
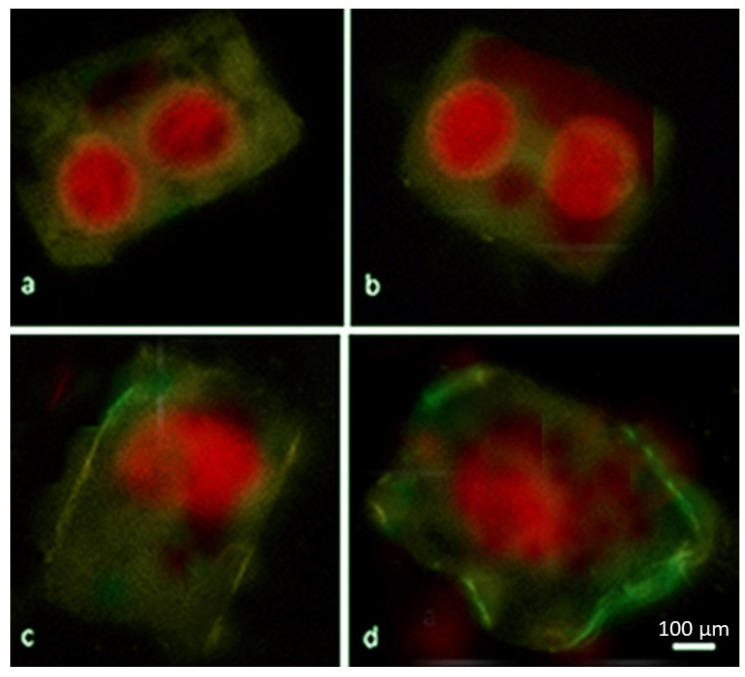
Localization of phosphatidylserine (green) on the surfaces of plasma membranes and propidium iodide in nuclei (red) of root cells of 6-day-old seedlings of *T. aestivum* variety Orenburgskaya 22 (**a**,**b**) and *T. durum* variety Zolotaya (**c**,**d**) after exposure to hypoxia for 24 h. Clusters of phosphatidylserine on plasmatic membrane surfaces (Annexin V-FITC). Bar: 100 μm.

**Figure 4 ijms-24-16878-f004:**
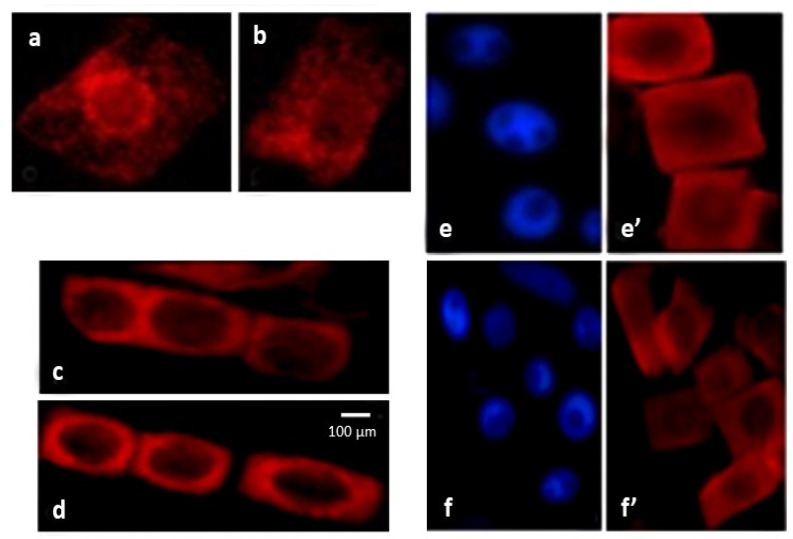
Immunodetection of cytochrome c in the cytoplasm of root cells of 6-day-old seedlings of *T. aestivum* variety Orenburgskaya 22 and *T. durum* variety Zolotaya. Cytochrome c in mitochondria of normal cells: (**a**) Orenburgskaya 22 and (**b**) Zolotaya; cytochrome c in the cytoplasm of cells in the presence of 150 mM NaCl: (**c**) Orenburgskaya 22 and (**d**) Zolotaya; nuclei (DAPI): (**e**) Orenburgskaya 22 and (**f**) Zolotaya; and cytochrome c in the cytoplasm of cells after exposure to hypoxia: (**e′**) Orenburgskaya 22 and (**f′**) Zolotaya. Bar: 100 μm.

**Figure 5 ijms-24-16878-f005:**
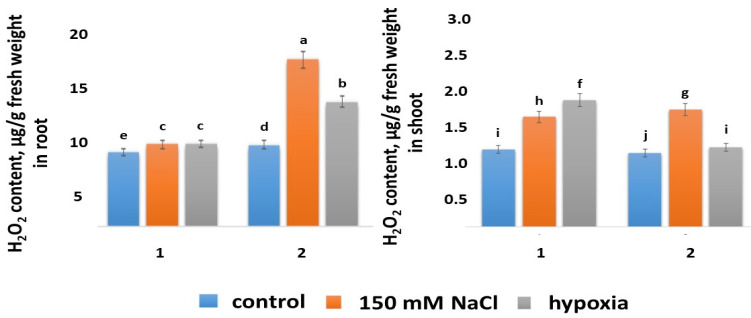
H_2_O_2_ contents in roots and shoots in 6-day seedlings of Orenburgskaya 22 variety (1) and Zolotaya variety (2). a–j—indicate significant difference were determined (*p* < 0.05).

**Figure 6 ijms-24-16878-f006:**
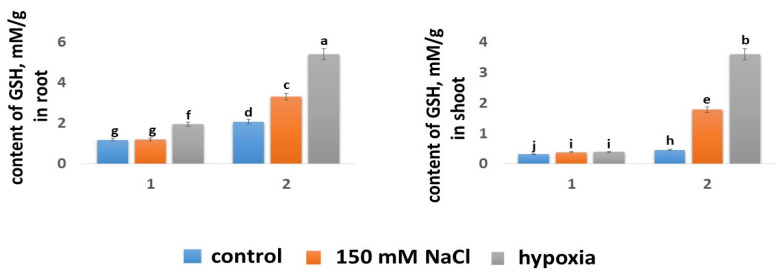
GSH contents in roots and shoots of 6-day seedlings of Orenburgskaya 22 variety (1) and Zolotaya variety (2). a–j—indicate significant difference were determined (*p* < 0.05).

**Figure 7 ijms-24-16878-f007:**
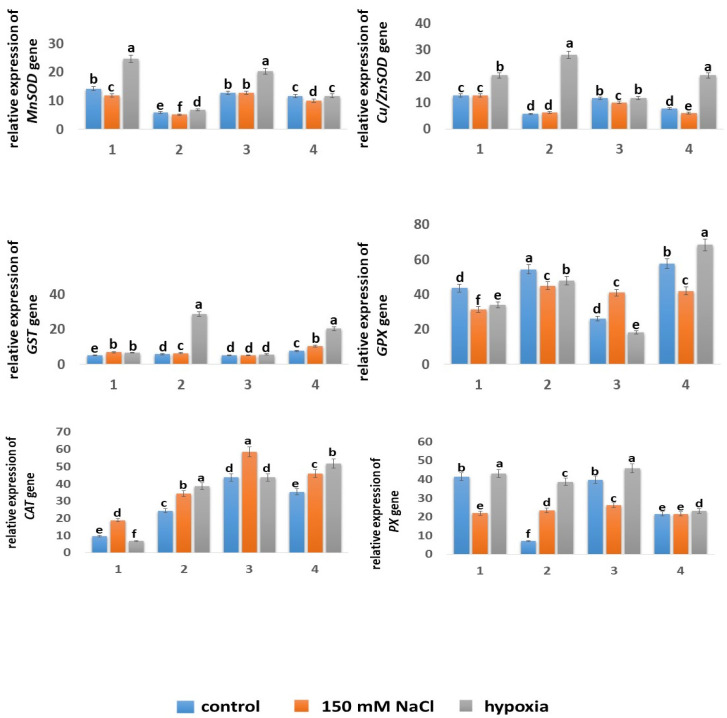
Expressions of the *MnSOD*, *Cu*/*ZnSOD*, *GPX*, *GST*, *CAT*, and *PX* genes in 6-day-old wheat seedlings of Orenburgskaya 22 (1, 3) and Zolotaya (2, 4) varieties under the influence of stress factors in 1, 2—roots and 3, 4—shoots. a–f—indicate significant difference were determined (*p* < 0.05).

**Figure 8 ijms-24-16878-f008:**
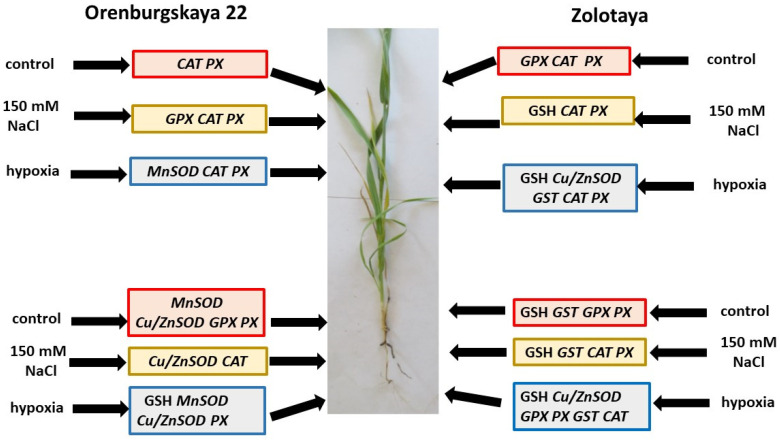
Scheme of AOS activation in roots and shoots of different wheat genotypes under the influence of stress factors.

**Table 1 ijms-24-16878-t001:** Morphometric parameters of 6-day seedlings of Orenburgskaya 22 and Zolotaya wheat varieties grown under different conditions.

Wheat Variety	Growth Condition	Seedling Length (cm)	Shoot Height (cm)	Root Length (cm)
Orenburgskaya 22	Control	28.9 ± 1.44 a	17.4 ± 0.87 a	12.5 ± 0.62 a
	150 mM NaCl	23.4 ± 1.17 c	12.3 ± 0.61 c	11.1 ± 0.55 b
	Hypoxia	27.6 ± 1.38 b	15.0 ± 0.75 b	12.6 ± 0.63 a
Zolotaya	Control	27.7 ± 1.38 a	16.0 ± 0.8 a	11.7 ± 0.58 b
	150 mM NaCl	19.4 ± 0.97 c	9.6 ± 0.48 c	9.8 ± 0.49 c
	Hypoxia	26.9 ± 1.34 b	13.2 ± 0.66 b	13.7 ± 0.68 a

Data were obtained from three repetitions. Data are expressed as means ± standard deviation (SD; n = 30), and a–c—indicate significant difference were determined (*p* < 0.05).

**Table 2 ijms-24-16878-t002:** Contents of Chl a and b in 6-day seedlings of Orenburgskaya 22 and Zolotaya wheat varieties grown under different conditions.

Wheat Variety	Treatment	Chl a (mg/g)	Chl b (mg/g)	Chl a/Chl b
Orenburgskaya 22	Control	4.13 ± 0.2 a	1.69 ± 0.08 a	2.44 ± 0.12 d
	150 mM NaCl	1.97 ± 0.1 d	0.71 ± 0.03 d	2.77 ± 0.14 c
	Hypoxia	1.25 ± 0.06 f	0.46 ± 0.02 e	2.72 ± 0.13 c
Zolotaya	Control	2.47 ± 0.12 c	0.88 ± 0.04 c	2.89 ± 0.14 a
	150 mM NaCl	1.34 ± 0.07 e	0.47 ± 0.02 e	2.82 ± 0.14 b
	Hypoxia	2.70 ± 0.13 b	0.92 ± 0.05 b	2.93 ± 0.15 a

Data were obtained from three repetitions. Data are expressed as means ± standard deviation (SD; n = 30), and a–f—indicate significant difference were determined (*p* < 0.05).

**Table 3 ijms-24-16878-t003:** Antiradical activity (ARA) and antioxidant activity (AOA) in the roots and shoots of 6-day seedlings of Orenburgskaya 22 and Zolotaya wheat varieties.

Wheat Variety	Growth Condition	Antiradical Activity, % (DPPH Method)	Antioxidant Activity, % (HMAEB Method)
Orenburgskaya 22	Control	56.86 ± 2.84 a	39.04 ± 1.95 a
Roots	150 mM NaCl	54.61 ± 2.73 b	36.94 ± 1.85 c
	Hypoxia	43.63 ± 2.18 g	25.98 ± 1.3 i
Orenburgskaya 22	Control	52.08 ± 2.6 d	36.15 ± 1.81 d
Shoots	150 mM NaCl	49.8 ± 2.49 e	30.5 ± 1.52 h
	Hypoxia	36.74 ± 1.84 h	23.24 ± 1.16 j
Zolotaya	Control	54.24 ± 2.71 b	35.86 ± 1.79 e
Roots	150 mM NaCl	34.38 ± 1.72 i	12.67 ± 0.13 l
	Hypoxia	49.57 ± 2.48 e	31.56 ± 1.58 g
Zolotaya	Control	53.42 ± 2.67 c	37.12 ± 1.85 b
Shoots	150 mM NaCl	31.24 ± 1.56 j	18.67 ± 0.93 k
	Hypoxia	47.05 ± 2.35 f	33.74 ± 1.69 f

Data were obtained from three repetitions. Data are expressed as means ± standard deviation (SD; n = 30), and a–l—indicate significant difference were determined (*p* < 0.05).

## Data Availability

Data are contained within the article and Appendix A.

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
