# Peer review of "Mechanisms of Antioxidant Resistance in Different Wheat Genotypes under Salt Stress and Hypoxia"

_ijms, 2023, doi:10.3390/ijms242316878_

Round 1

Reviewer 1 Report (New Reviewer)

Comments and Suggestions for Authors

The authors have analyzed the formation and localization of ROS in two wheat varieties under saline and hypoxia stress. In general, plants have a powerful antioxidant system (AOS) to neutralize excess ROS. Through the results authors revealed that, two wheat varieties, Orenburgskaya 22 and Zolotaya, showed different resistance to abiotic stresses. Orenburgskaya 22 showed high resistance to salt stress but sensitivity to hypoxia, while Zolotaya showed tolerance to hypoxia but high sensitivity to salt stress. The Zolotaya variety's tolerance to hypoxia is due to glutathione content and enzyme activation. Understanding plant tolerance mechanisms is crucial for improving agricultural plant breeding and yield. The reviewer appreciates the effort of the authors to prove their hypothesis using series experiments. However, the reviewer has a few comments regarding this study. Thus, the authors need to consider the following comments to improve the quality of this manuscript.

Line 12: Plant scientific names should be in full form (both genus and species) in first mention rest should be abbreviated format (T. durum). Authors should revise this throughout the manuscript including figure legends.

Line 96-99: Needs to be improved with more detail.

Authors are advised to fix grammar, space, punctuation, and formatting errors throughout the manuscript.

Please reduce the usage of ‘We’ in the MS. Eg. In this study focused on…

Please insert the title of ‘results’ section in the MS.

Write the meaning of different letters mentioned in the Table 1 footnotes. Follow the same in all the table footnotes.

All figures quality is not good. Enhance it.

Line 197, 213, 225: Bar 100 µm should be ‘scale bar represents 100 µm’.

Line 234: hypoxia à Hypoxia

Line 249: in vitro should be italics.

Please describe more about in the RNA extraction and RT-PCR section in the materials and method section. What is RNA concentration? How much concentration was used for cDNA conversion. Cite the appropriate reference.

Concise the conclusion section. 

Author Response

We are grateful to you for your kind review, valuable comments and recommendations.

Reviewer 2 Report (New Reviewer)

Comments and Suggestions for Authors

In this manuscript, authors investigated the mechanisms of antioxidant resistance of two wheat varieties, Orenburgskaya 22 and Zolotaya under two treatments of 150 mM NaCl and hypoxia. This study could be important for understanding the genetic mechanism of salt stress and hypoxia. However, the manuscript does have several issues that need to be addressed.

Materials and Methods

The description of Materials and Methods should be improved. Authors need to provide reasons for choosing the two wheat varieties, Orenburgskaya 22 and Zolotaya for the study. The experiment design is unclear. What type of design did authors use, and how many replications were there for each genotype per treatment?

1.      line 57-58, Change “cell death (PCD)” to “programmed cell death (PCD)”.

2.      line 91, spell out AOS, “antioxidant system (AOS)".

3.      Line 96-97, Why did authors choose the two varieties for the study?

4.      line 100, The subtitle “2. Results” is missing.

5.      Table 1 is unclear. The two treatments, 150 mM NaCl and hypoxia should be analyzed separately. Additionally, since the two variety controls were significantly different before treatments began, how did authors make a meaningful comparison for the difference due to treatment, or genotype? Authors should use relative difference to the control for the statistical analysis. Regarding the hypoxia treatment, the relative shoot height of Orenburgskaya 22 (17.4-15=2.4 cm) and Zolotaya (16-13.2=2.8 cm) is small. Is it is statistically different?

6.      Line 299, “in Figure 7A and B.” however, Figure 7 does not have A and B. Again, authors need to analyze the two treatments, two tissues separately.

7.      Figure 8, authors may need to use two plants instead of one to represent two different varieties.

8.      Line 639, "150 mm NaCl" should be "150 mM NaCl"  

Author Response

We are grateful to you for your kind review, valuable comments and recommendations.

Round 2

Reviewer 2 Report (New Reviewer)

Comments and Suggestions for Authors

Thank authors for respond to my questions. However, authors do not answer my question#3 and #5 clearly.

 Authors need to provide evidence to support their argument that two wheat varieties turned out to be excellent models.

I hope authors answer questions point by point for question#5. For instance, the two controls were significantly different before treatments. How do the authors know the difference is due to the treatment since the two controls of varieties were significantly different before treatments? Therefore, the relative difference should be used for statistical analysis in order to have meaningful comparison.

Author Response

Thanks for your comments. We apologize for the incomplete answers.
3. We have carefully revised the entire text, checked and corrected grammar, spaces and formatted the text
5. Thanks for your comment. We agreed with you and conducted statistical analysis separately for each variety. When using a statistically significant value of p < 0.01, the values fit into our table, but for p < 0.05 it is really better to report the statistics separately for each variety. Indeed, the two control wheat varieties differed before treatment, we only wanted to indicate the trend of changes during treatment and compared the relative difference in their changes
These two wheat varieties are different genotypes and have become models for studying the mechanisms of adaptation to different types of stress. As a result of our study, different adaptation mechanisms were identified in different wheat genotypes.

Round 3

Reviewer 2 Report (New Reviewer)

Comments and Suggestions for Authors

Authors addressed my comments in the manuscript. I think the paper is now acceptable for publication.

This manuscript is a resubmission of an earlier submission. The following is a list of the peer review reports and author responses from that submission.

Round 1

Reviewer 1 Report

Comments and Suggestions for Authors

The recent manuscript is based on at least three previously published papers from the authors with the new aim, to investigate not only the restorative, but the protective effects of quercetin. The manuscript is written in good manners and order. The authors did a lot of efforts to prove their hypothesis, still need to answer the raised questions and improve/correct the manuscript before publication:

- Why the authors did not put the results of the restorative effects of quercetin on ROS level, only the results of the protective effects of quercetin on ROS level?

- Write in paragraph 4.1 how hypoxia was exatly induced.

- How the quercetin solution was prepared? Did you use organic solvent? In that case solvent control is also necessary during the whole experiment, as spraying with for example diluted ethanol can change the antioxidant system of the plants.

- How the investigated genes were selected? There are many PX, GST or GPX genes in the wheat? Why did you select that exact one? Need a good explanation for that. Also indicate with an access code/number that which one is the investigated gene (in every case). This way is random and chaotic.

-There are a lot of random formatting mistakes throughout the manuscript (bold or not bold texts). I don't know that it happened during the Pdf conversion or before, but need to correct.

Author Response

We are grateful for your comments

The recent manuscript is based on at least three previously published papers from the authors with the new aim, to investigate not only the restorative, but the protective effects of quercetin. The manuscript is written in good manners and order. The authors did a lot of efforts to prove their hypothesis, still need to answer the raised questions and improve/correct the manuscript before publication:

- Why the authors did not put the results of the restorative effects of quercetin on ROS level, only the results of the protective effects of quercetin on ROS level?

Reply: The experiment was carried out in full, data on ROS were obtained for two ways of A/B treatment with quercetin. Quercetin has only a partially restorative effect only on the Orenburgskaya 22 variety, and then only in the case of salt exposure. Therefore, we did not present these data in the manuscript, as we wanted to discuss them in more detail later.

- Write in paragraph 4.1 how hypoxia was exactly induced.

Reply: All experiments were carried out on a roll culture, so it was convenient for us to conduct an experiment with flooding, bringing the water level to the hypocotyl. We have included a more detailed description of the hypoxia experience.

- How the quercetin solution was prepared? Did you use organic solvent? In that case solvent control is also necessary during the whole experiment, as spraying with for example diluted ethanol can change the antioxidant system of the plants.

Reply: Thank you for your question, we have added the preparation of a quercetin solution to the method. Quercetin was dissolved in DMSO to obtain a 3% stock solution, then diluted with water to a concentration of 0.3%. The DMSO solution was analyzed for ROS. It had no effect on the change in ROS.

- How the investigated genes were selected? There are many PX, GST or GPX genes in the wheat? Why did you select that exact one? Need a good explanation for that. Also indicate with an access code/number that which one is the investigated gene (in every case). This way is random and chaotic.

Reply: Thanks for the question. We agree that there are many PX, GST or GPX genes in wheat. And although in the study we also used other genes of these families, we presented data from these genes, which in our opinion had more effective meanings. We have added the accession numbers of the genes used to the Table 1S.

-There are a lot of random formatting mistakes throughout the manuscript (bold or not bold texts). I don't know that it happened during the Pdf conversion or before, but need to correct.

Reply: Sorry, there was a problem entering text into the journal format; there were none in the original version. We corrected everything

Reviewer 2 Report

Comments and Suggestions for Authors

MS is poorly written and is often hard to understand. The MS is full of typos, syntax and grammatical errors. Acronyms need to be expanded at first mention. Scientific names should be provided at first mention. MS is unstructured and unorganized. A clear rationale/hypothesis is missing. Conclusions are vague. A few examples of very bad writing are; L12 Meristema; L20 ; different activation of antioxidant system; L21; The basis of the protective reaction of the AOS variety Zolotaya; L105 morphmetric; L244 ‘Effect quercetin on ROS’

L24; Treatment of wheat seedlings with antioxidant quercetin has a protective effect, preventing the toxic effects of abiotic factors’. Which variety of wheat seedlings?

L25 level of expression genes in wheat cells; which genes and which variety and which cells

L28 ‘This study made it possible to identify effective AOS components that neutralize the negative effect of ROS in different wheat genotypes.’ The authors claim to have identified the responsible component but I do not see it

In the introduction section, the Authors explain the importance of ‘hypoxia’ but not the salt stress/salinity. Please add some lines and link both hypoxia and salt to ROS.

Why did the authors choose to expose plants to Salt or hypoxia stress specifically? A clear rationale is missing

The rationale for selecting these two specific wheat varieties needs to be mentioned

L108: ‘1% H2O2 as a model for the additional formation of ROS in wheat.’ What do you mean by model for additional ROS production?

It will be better to convert table datasets (Table 1, 2, 3) to graphs for better visualization and inference.

<ore detailed legends for figures are needed

Pictures provided in Figure 1 (A and B) are not clear and one can not make out what and where to focus on Figure 1 to interpret the results

It is a very strange representation of data; why did you split Figure 1 into 1A and 1B with two separate legends?

I assume the relative fluorescence depicted in Figure 2 is derived from your Figure 1. In that case it does not qualify for a separate standalone figure. Figure 2 thus, should be merged with Figure 1.

The quality of figures 2-4 is very bad. Markings are lacking in the figure to emphasize what authors wan t o say.

From Figure 3, how can one differentiate between normal and damaged DNA? The cell size in each panel is different. I have the same comment for all the figures.

Which antiradical and antioxidant activities were measured for Table 3?

Statistics are missing from all the graphs.

Comments on the Quality of English Language

MS is poorly written and is often hard to understand. The MS is full of typos, syntax and grammatical errors. A few examples of very bad writing are; L12 Meristema; L20 ; different activation of antioxidant system; L21; The basis of the protective reaction of the AOS variety Zolotaya; L105 morphmetric; L244 ‘Effect quercetin on ROS’

Author Response

We are grateful for your comments

MS is poorly written and is often hard to understand. The MS is full of typos, syntax and grammatical errors. Acronyms need to be expanded at first mention. Scientific names should be provided at first mention. MS is unstructured and unorganized. A clear rationale/hypothesis is missing. Conclusions are vague. A few examples of very bad writing are; L12 Meristema; L20 ; different activation of antioxidant system; L21; The basis of the protective reaction of the AOS variety Zolotaya; L105 morphmetric; L244 ‘Effect quercetin on ROS’

L24; Treatment of wheat seedlings with antioxidant quercetin has a protective effect, preventing the toxic effects of abiotic factors’. Which variety of wheat seedlings?

Reply: Sorry, this was a general phrase and did not apply to the results of this study. We rewrote the abstract

L25 level of expression genes in wheat cells; which genes and which variety and which cells

Reply: Sorry, this was a general phrase and did not apply to the results of this study. We rewrote the abstract

L28 ‘This study made it possible to identify effective AOS components that neutralize the negative effect of ROS in different wheat genotypes.’ The authors claim to have identified the responsible component but I do not see it

Reply: We are only talking about identifying the effective components of the AOS system and identifying the difference in the most effective AOS components in different wheat varieties under the influence of salt stress and hypoxia. We have clarified this proposal in the abstract

In the introduction section, the Authors explain the importance of ‘hypoxia’ but not the salt stress/salinity. Please add some lines and link both hypoxia and salt to ROS.

Reply: Thank you for your comment. Although we previously noted the importance of studying salt stress, we have added an addition to the introduction

Why did the authors choose to expose plants to Salt or hypoxia stress specifically? A clear rationale is missing

Reply: These two stress factors are most typical for mid-latitudes.

The rationale for selecting these two specific wheat varieties needs to be mentioned

Reply: We analyzed different varieties of wheat. These two varieties showed the most striking differences in response to salt stress and hypoxia

L108: ‘1% H2O2 as a model for the additional formation of ROS in wheat.’ What do you mean by model for additional ROS production?

Reply: Sorry for the incorrect use of “model” in this sentence. Corrected to “Treatment of wheat with 1% H2O2 has been considered to increase ROS production.”

It will be better to convert table datasets (Table 1, 2, 3) to graphs for better visualization and inference.

Reply: Thanks for the offer. We still decided to leave the tables, since presenting only graphs in the manuscript, in our opinion, worsens the design

<ore detailed legends for figures are needed

Reply: Thanks for the offer. We tried to describe the legends to the pictures and graphs in more detail.

Pictures provided in Figure 1 (A and B) are not clear and one can not make out what and where to focus on Figure 1 to interpret the results

Reply: We tried to improve the visualization of Figure 1. Attention should be paid to the zones of the root with ROS and the intensity of fluorescence

It is a very strange representation of data; why did you split Figure 1 into 1A and 1B with two separate legends?

I assume the relative fluorescence depicted in Figure 2 is derived from your Figure 1. In that case it does not qualify for a separate standalone figure. Figure 2 thus, should be merged with Figure 1.

Reply: Thanks for the offer. We have combined Figures 1A and B and Figure 2 and combined their legends.

The quality of figures 2-4 is very bad. Markings are lacking in the figure to emphasize what authors wan t o say.

Reply: Thank you for your note, we have changed the markings of the figures

From Figure 3, how can one differentiate between normal and damaged DNA? The cell size in each panel is different. I have the same comment for all the figures.

Reply: Thanks for the question. Nuclei with intact DNA can be distinguished from nuclei with DNA breaks by the presence of green staining in the area of the nuclei

Which antiradical and antioxidant activities were measured for Table 3?

Reply: These two methods determine the antioxidant activity of drugs. However, the determination of antioxidant activity using DPPH is also called antiradical activity, since DPPH is a chromogen radical. Activities were measured as % inhibition of oxidative reaction

Statistics are missing from all the graphs.

Reply: Thank you for your comment. We have added statistics to all graphs

Reviewer 3 Report

Comments and Suggestions for Authors

The manuscript "Mechanisms of antioxidant resistance in different wheat genotypes" by Kononenko et al. is a study on the interaction between ROS and antioxidant in the response of two wheat varieties on salt stress and hypoxia. The novelty of this paper lies in the effect of the treatment with quercetin on plants under these different kinds of stresses. The results were collected properly and obviously a lot of work went into the experimental work. I therefore support that these data will eventually be published.

It is on the other hand also clear, as is the case with many stress metabolism papers, that the story is drowned in the many combinations of treatments and varieties. The main message is lost in a wide range of tables and data and in the end it is not clear what the paper really wants to show and how it distinguishes itself from other papers by the same authors. That is, there is important information to be found and presented, but in the end, a lot of individual observations are made, and we do not gain real new insights.

For example, what is not clear to me is why the authors insist on giving data on both wheat varieties, instead in going into depth on one of them. They do tend to make some comparisons but never flesh out the details. There are too many good comparisons to be made, so pick the new ones and stick to it. The authors start out with their plan A/B story (quercetin before/after the treatment) but never come back to it. Nevertheless, this is a very new idea and it is sad that in the end it is lost among a lot of observations and in a discussion that ultimately rehashes concepts that were discussed before.

And this shows in the title already. It is too broad, too vague, and does not play to the strength of the dataset.

The authors may remedy this:

- by providing an overview table of the observed effects at the end of the paper, as a guideline for the discussion.

- by working on the statistics and adding linear models (see below, Methodology)

- An alternative could also be that the discussion is restructured. The authors should mix results and discussion, and write the paper from a hypothesis-driven point, setting up new hypotheses with each new piece of evidence. This would make the paper much easier to follow.

- by focusing on what makes this paper really new.

Methodology:

In the part about the statistical methods, the authors should explain how they corrected for multiple testing. They should also perform two-way ANOVA on their data, to clarify individual and combined effects of variety, and quercetin treatment. Differences between stress treatments seem irrelevant to me, as it only complicates the analysis and franky, it is comparing apples and oranges.

In Tables 1/2: do letters also include differences between the varieties ?

Discussion

Line 508: refs 76-78 do not talk about PCD (the authors should add a proper reference, eg 68). It is also intriguing that the authors do mention changes in metabolism, yet regularly call the stress "toxic" (is that so?) and even use lower levels of H2O2 after treatment a sign of resistance against salinity. I would recommend the use of the word tolerance, and the authors should consider in their discussion that ROS production may at least be a form of signal generation leading to this metabolic rearrangement. Of course, on line 530, the authors return to the PCD option as an alternative outcome of stress exposure. Should this then be considered distress or is this a way to handle eustress after all (see Lichtenthaler 1996, 1998), and is it then a necessary part of this rearrangement?

The authors should back up all their statements with referral to their own figures.

Minor comments:

- Figure 1 AB : add tjhe meaning of EZ, DZ, MZ to the caption

- Figure 2: no significances have been indicated on the figure. ALso, based on this fig ther seems to be a significant effect of quercetin in Zolotaya, but the authors say the contrary on lines 592-594

- Also no significances on fig 6-7-8

- paragraph lines 575-585 should be integrated and supported with data from the study itself because now it rehashes things we know already. Also, just relying on chlorophyll ratios is not enough to talk about photosynthesis redox signalling.

Some references:

Lichtenthaler, H.K. (1996). Vegetation Stress: an introduction to the stress concept in plants. Journal of Plant Physiology 148: 4-14.

Lichtenthaler, H.K. (1998). The stress concept in plants: an introduction. Annals of the New York Academy of Sciences 851(1): 187-198.

Comments on the Quality of English Language

The paper should be re-read for spelling mistakes, especially in the figures/tables:

- hEIght in table 1

- hYpoxia in figure

Author Response

Thank you for your valuable comments and recommendations

The manuscript "Mechanisms of antioxidant resistance in different wheat genotypes" by Kononenko et al. is a study on the interaction between ROS and antioxidant in the response of two wheat varieties on salt stress and hypoxia. The novelty of this paper lies in the effect of the treatment with quercetin on plants under these different kinds of stresses. The results were collected properly and obviously a lot of work went into the experimental work. I therefore support that these data will eventually be published.

It is on the other hand also clear, as is the case with many stress metabolism papers, that the story is drowned in the many combinations of treatments and varieties. The main message is lost in a wide range of tables and data and in the end it is not clear what the paper really wants to show and how it distinguishes itself from other papers by the same authors. That is, there is important information to be found and presented, but in the end, a lot of individual observations are made, and we do not gain real new insights.

Reply: Thanks for your critical comments. To solve the problems and prove the results obtained, it was necessary to conduct many different experiments. We agree that not all the results obtained, unfortunately, were properly discussed; we only concentrated on one idea, otherwise the size of the manuscript could have increased significantly.

For example, what is not clear to me is why the authors insist on giving data on both wheat varieties, instead in going into depth on one of them. They do tend to make some comparisons but never flesh out the details. There are too many good comparisons to be made, so pick the new ones and stick to it. The authors start out with their plan A/B story (quercetin before/after the treatment) but never come back to it. Nevertheless, this is a very new idea and it is sad that in the end it is lost among a lot of observations and in a discussion that ultimately rehashes concepts that were discussed before.

Reply: Thank you for your comments and suggestions. We agree that we focused on one idea and we used the A/B path only as a tool to prove that idea. We plan to develop and discuss many interesting facts in more detail in our further research.

And this shows in the title already. It is too broad, too vague, and does not play to the strength of the dataset.

Reply: Thank you for your comment. We agree that this name is too broad and we have changed it slightly.

The authors may remedy this:

- by providing an overview table of the observed effects at the end of the paper, as a guideline for the discussion.

Reply: Thanks for the offer. We considered that the summary diagram was sufficient, since our overview table turned out to be huge.

- by working on the statistics and adding linear models (see below, Methodology)

Reply: Thank you, we have done some work with statistics

- An alternative could also be that the discussion is restructured. The authors should mix results and discussion, and write the paper from a hypothesis-driven point, setting up new hypotheses with each new piece of evidence. This would make the paper much easier to follow.

Reply: Thanks for the offer. We tried to rework the Discussion.

- by focusing on what makes this paper really new.

Methodology:

In the part about the statistical methods, the authors should explain how they corrected for multiple testing. They should also perform two-way ANOVA on their data, to clarify individual and combined effects of variety, and quercetin treatment. Differences between stress treatments seem irrelevant to me, as it only complicates the analysis and franky, it is comparing apples and oranges.

In Tables 1/2: do letters also include differences between the varieties ?

Reply: Statistical data are presented only within the variety and within different routes of quercetin treatment.

Discussion

Line 508: refs 76-78 do not talk about PCD (the authors should add a proper reference, eg 68). It is also intriguing that the authors do mention changes in metabolism, yet regularly call the stress "toxic" (is that so?) and even use lower levels of H2O2 after treatment a sign of resistance against salinity. I would recommend the use of the word tolerance, and the authors should consider in their discussion that ROS production may at least be a form of signal generation leading to this metabolic rearrangement. Of course, on line 530, the authors return to the PCD option as an alternative outcome of stress exposure. Should this then be considered distress or is this a way to handle eustress after all (see Lichtenthaler 1996, 1998), and is it then a necessary part of this rearrangement?

The authors should back up all their statements with referral to their own figures.

Reply: We have supplemented the Discussion section with our digital data

Reply: Thanks for the comment and recommendations. We tried to edit the discussion section with your recommendations and added a reference

Minor comments:

- Figure 1 AB : add tjhe meaning of EZ, DZ, MZ to the caption

Reply: Added to the legend of Figure 1

- Figure 2: no significances have been indicated on the figure. ALso, based on this fig ther seems to be a significant effect of quercetin in Zolotaya, but the authors say the contrary on lines 592-594

Reply: Figure 1C(2) shows data only for quercetin pretreatment (pathway B). In the Discussion section, we say that pre-treatment with quercetin leads to a protective effect in both wheat varieties, especially the Orenburgskaya 22 variety, and reduces the intensity of ROS, as confirmed by the data in Figure 1C(2).

- Also no significances on fig 6-7-8

Reply: Thank you for your note, we have added alphabetic statistics to the drawings

- paragraph lines 575-585 should be integrated and supported with data from the study itself because now it rehashes things we know already. Also, just relying on chlorophyll ratios is not enough to talk about photosynthesis redox signalling.

Reply: We agree with you, however, the chlorophyll ratio can be used as one of the markers of redox status

Some references:

Lichtenthaler, H.K. (1996). Vegetation Stress: an introduction to the stress concept in plants. Journal of Plant Physiology 148: 4-14.

Lichtenthaler, H.K. (1998). The stress concept in plants: an introduction. Annals of the New York Academy of Sciences 851(1): 187-198.

Reply: Thanks for the references. We used them and added them to the references of our manuscript

Reviewer 4 Report

Comments and Suggestions for Authors

In the manuscript the authors aimed to study the formation and localization of ROS in wheat Triticum aestivum and Triticum durum as a result of exposure to abiotic stresses such as saline and hypoxia, to identify the active components of AOS and to determine the effectiveness of the antioxidant quercetin for removing ROS under the action of abiotic stresses. The manuscript is rich of data and of literature supporting the results, making it exhaustive and effective for obtaining the objectives declared. But just because of the rich literature existing on the topic and because of similar results and experiments already published, I found the work with low originality and novelty. Anyway, I think that some important points need to be better explained and clarified.

The major comments are about the methods section that needs to be more detailed:

·        Better explain the reasons of the choice of the two wheat varieties object of the experiment: did you had any previous information supporting this choice (different tolerance to abiotic stresses)?

·        How hypoxia condition was obtained in the experiment? Please describe the methods in more detail.

·        In section 4.6 the method for cytochrome c detection is not described, and no reference is reported.

Results section

·        Section 2.5.4: how the expression of the genes was measured? Which measuring unit is reported in the Y axis of Figures 8A and B?

Conclusion

·        I found the conclusion a synthetic repetition of results and discussion. Conclusion should focus on the future perspectives, on the importance and on potential use of the results obtained.

Author Response

We are grateful for your kind comments and remarks.

In the manuscript the authors aimed to study the formation and localization of ROS in wheat Triticum aestivum and Triticum durum as a result of exposure to abiotic stresses such as saline and hypoxia, to identify the active components of AOS and to determine the effectiveness of the antioxidant quercetin for removing ROS under the action of abiotic stresses. The manuscript is rich of data and of literature supporting the results, making it exhaustive and effective for obtaining the objectives declared. But just because of the rich literature existing on the topic and because of similar results and experiments already published, I found the work with low originality and novelty. Anyway, I think that some important points need to be better explained and clarified.

The major comments are about the methods section that needs to be more detailed:

  • Better explain the reasons of the choice of the two wheat varieties object of the experiment: did you had any previous information supporting this choice (different tolerance to abiotic stresses)?

Reply: Thank you for your comment. We conducted an experiment with several varieties of wheat and one of the varieties (Orenburgskaya 22) showed tolerance in some parameters to salt stress, and the Zolotaya variety showed tolerance to hypoxia. On this basis, they were selected for a more detailed study.

  • How hypoxia condition was obtained in the experiment? Please describe the methods in more detail.

Reply: Thank you for your comment. We described the experience with hypoxia in more detail.

  • In section 4.6 the method for cytochrome c detection is not described, and no reference is reported.

Reply: Thank you for your comment. We described the experience with hypoxia in more detail.

Results section

  • Section 2.5.4: how the expression of the genes was measured? Which measuring unit is reported in the Y axis of Figures 8A and B?

Reply: Sorry for the mistake, of course the data is relative gene expression. We have made changes to Figure 8A and B.

Conclusion

  • I found the conclusion a synthetic repetition of results and discussion. Conclusion should focus on the future perspectives, on the importance and on potential use of the results obtained.

Reply: Thank you for your recommendations, we have revised the conclusion according to your commen

Round 2

Reviewer 1 Report

Comments and Suggestions for Authors

The authors answered the raised questions and corrected the manuscript according to the suggestions, except for one point. However, they mentioned in their answer but did not put the quercetin solution preparation in the methods. It is probably just missed, but please, add it. Otherwise in my opinion the quality of the research work and the manuscript is adequate for publication.

Reviewer 3 Report

Comments and Suggestions for Authors

I have re-read the manuscript by Kononenko et al. on antioxidant resistance of two wheat varieties towards salt stress and hypoxia.

I still cannot recommend publication of this manuscript, for these four reasons:

1. The dataset is still too complex with a set of conditions at the beginning of which the significance is lost halfway through the text, like the A/B testing phase with quercetin. The authors note in their reply that they did concentrate on their idea, but still show all the data which makes it very difficult to get the point. I also understand their argument that treating all the data and the treatments would increase the length of the text too much, but this goes against clarity.

2. To improve clarity and make sense of the many treatments, I had suggested to do two-way ANOVA analyses on the data. The authors apparently did not read that part. I had also asked to add how they took care of multiple testing in their many series of t-tests. There was no answer. I therefore do not trust the statistics anymore.

3. In their rebuttal, the authors state that they tried to rework the Discussion. Apart from the insertion of the references to their own figures and one reference to Lichtenthaler (indeed, at my suggestion), NOTHING has changed. To quote a great philosopher: "Do, or do not. There is no try."

4. The authors did not follow up on a proper discussion on the nature of their stresses (eustress/distress). This is crucial to understand the effect of their treatments.

Comments on the Quality of English Language

There are some typo's or typical Slavic problems with the definite article. Nothing a proper reread by someone with English as mother tongue can't fix

(e.g in the title under salt stress and hypoxia, not THE salt stress)

Reviewer 4 Report

Comments and Suggestions for Authors

The authors not fully addressed my previous concerns. Furthermore, the experimental design and the method section still contain important critical issues: some measurement were conducted in 10 days years old seedling, others in 4 days years old seedlings. Which parameters were measured in root tissues, which in shoot tissues?

The results section and discussion section are not clearly separated in term of contents: result section contains discussion and references to other works, discussion section contains results and reference to figures and tables.

Authors many times in the manuscript refers to “response to various abiotic stresses”, but plant response to only salt and hypoxia stress was measured.

Conclusions are generic and poorly related with results and discussions and mainly related with literature cited.

For these reasons the manuscript cannot be accepted in the present form.